# Insights on Shear Transfer Efficiency in “Brick-and-Mortar” Composites Made of 2D Carbon Nanoparticles

**DOI:** 10.3390/nano12081359

**Published:** 2022-04-15

**Authors:** Fabrizia Cilento, Alfonso Martone, Michele Giordano

**Affiliations:** 1Department of Chemical, Materials and Production Engineering, University of Naples Federico II, 80125 Naples, Italy; fabrizia.cilento@unina.it; 2Institute of Polymers, Composite and Biomaterials (IPCB), National Research Council of Italy, 80055 Portici, Italy; michele.giordano@cnr.it; 3IMAST S.c.ar.l.—Technological District on Engineering of Polymeric and Composite Materials and Structures, 80133 Naples, Italy

**Keywords:** GNP, nanolaminates, brick-and-mortar

## Abstract

Achieving high mechanical performances in nanocomposites reinforced with lamellar fillers has been a great challenge in the last decade. Many efforts have been made to fabricate synthetic materials whose properties resemble those of the reinforcement. To achieve this, special architectures have been considered mimicking existing materials, such as nacre. However, achieving the desired performances is challenging since the mechanical response of the material is influenced by many factors, such as the filler content, the matrix molecular mobility and the compatibility between the two phases. Most importantly, the properties of a macroscopic bulk material strongly depend on the interaction at atomic levels and on their synergetic effect. In particular, the formation of highly-ordered brick-and-mortar structures depends on the interaction forces between the two phases. Consequently, poor mechanical performances of the material are associated with interface issues and low stress transfer from the matrix to the nanoparticles. Therefore, improvement of the interface at the chemical level enhances the mechanical response of the material. The purpose of this review is to give insight into the stress transfer mechanism in high filler content composites reinforced with 2D carbon nanoparticles and to describe the parameters that influence the efficiency of stress transfer and the strategies to improve it.

## 1. Introduction

In the last years, many efforts have been conducted to fabricate nanocomposites with high performances suitable for specific applications in different fields. In particular, the main challenge is to reproduce on the macroscale the mechanical and functional properties of the nanometric reinforcement. In nature, it is possible to find materials, such as bones or mollusc shells, that thanks to their well-organized hierarchical structure exhibit impressive performances. In particular, the inner part of the mollusc shell, called nacre, is characterized by a brick-and-mortar (B&M) structure, constituted by well-oriented thin laminae of aragonite bonded together by a small amount of organic material. Therefore, nacre can be assimilated to a composite material, capable of reproducing the mechanical properties of the lamellar reinforcement to the macroscale [1,2] and dissipating energy with localized plastic deformations, without experiencing global failure [3]. Accordingly, a new class of biomimetic materials, called nacre-like materials, which mimic the brick-and-mortar (B&M) architecture of nacre has been developed. These artificial nanolaminates are constituted by a high quantity of stiff but brittle nanoparticles, bonded together by a small amount of soft but tough phase.

In the following sections, a review on nacre-like composite materials is reported, highlighting fabrication methods employed and raw materials used and comparing structural properties.

### 1.1. Mimicking Nature: Nacre-Inspired Materials

Among all biomimetic materials, nacre has drawn great attention from the scientific community, thanks to superior levels of strength and toughness and its brick-and-mortar (B&M) architecture. Nacre is the iridescent inner shell layer of some molluscs. It consists of a 3D assembly of hard lamellar aragonite tablets glued together with a low amount (5 vol%) of soft organic materials (proteins and polysaccharides). This hierarchically organized microstructure and the small fraction of biopolymers are responsible for the unique mechanical behaviour. In addition, the mineral bridges that connect the different tiles at the nanoscale level are capable of preventing crack extension and providing toughness and impact resistance [4,5]. Although nacre is composed of fragile material, it exhibits a ductile behaviour, allowing plastic deformations. Its peculiarity is the high toughness, which is three orders of magnitude higher than its main constituents. Nevertheless, it is associated with different mechanisms acting at the nanoscale: (i) nanoasperities of the aragonites tiles; (ii) organic layer acting as viscoelastic glue after the elongation of biopolymer; (iii) mineral bridge relocking after fracture; (iv) tile interlock due to the microscale waviness and dovetail of tiles [4].

### 1.2. Engineering Materials Based on 2D Nanoparticles

Inspired by nacre, several synthetic materials with brick-and-mortar structure. have gathered the attention of scientists worldwide. In particular, several systems have been studied and various platelet/polymer structures have been investigated. Particular attention has been devoted to paper-like materials reinforced with lamellar fillers, which can reproduce on the macroscopic scale the mechanical characteristics of the nanoscale reinforcement. The mechanical and functional properties of this class of material can be designed according to the filler and matrix nature [6], as described in Table 1. The mechanical performances in terms of strength and stiffness are regulated by the filler properties, while the energy dissipation is regulated by the matrix brittle or ductile behaviour. However, the overall behaviour depends on the quality of stress transfer between the two phases and on their interactions. On the other hand, thermal and electrical properties depend only on the nature of the filler, due to the low amount of polymer, and are characterized by high anisotropy between in-plane and cross-plane conductivities [7,8].

In the literature, several attempts have been made to mimic and design artificial nacre. For this purpose, different materials have been employed as bricks and mortar. The most employed nanoparticles used as reinforcement are listed in Table 2, and a comparison in terms of costs, mechanical properties and thermal and electrical conductivities is presented.

A huge number of papers based on graphitic nanoparticles can be found in the literature. After its discovery in 2004 by the scientists Geim and Novoselov, graphene has been in the spotlight involving many researchers for possible applications in several industrial sectors thanks to its outstanding electronic, optical, thermal and mechanical properties. The sub-nanometric thickness of carbon–carbon bonding makes graphene the strongest material in the world, stronger than steel and Kevlar, with a tensile strength of 130 GPa and a Young’s modulus of 1 TPa [9]. It is elastic, returning to initial dimensions after the stretching [19], lightweight (0.77 mg/m^2^) [20] and able to absorb light [21]. Those characteristics make the material very attractive for scientific research and industrial applications. Despite all this, these properties refer to an ideal material, because producing monolayer graphene sheets without defects is extremely expensive and challenging. For this reason, other graphitic particles with similar characteristics are widely employed in the research field. They are graphene oxide (GO), reduced graphene oxide (RGO) and graphite nanoplatelets (GNPs). They differ from graphene by physical and chemical features such as average lateral size, number of layers and carbon/oxygen ratio (C/O) [22]. In fact, by increasing the number of layers of graphene, i.e., the thickness, or by decreasing the C/O ratio, both mechanical properties and conductivity of the material are negatively affected [23]. Consequently, the price of graphene is linked to its quality and to the technique used for its production. Mechanically exfoliated graphene (obtained with the “scotch tape” technique [24]) comes in small, high-quality flakes, not nearly enough for industrial applications, with a price of the order of several thousand euros per flake. However, GO and GNPs, produced by oxidation and exfoliation of graphite, can be mass-produced, cutting the costs to tens of euros per gram.

In addition, nanoclays are widely used to fabricate nacre-like composites, thanks to their low costs (<1 EUR/g) combined with remarkable mechanical properties. These are a broad class of natural inorganic minerals, of which montmorillonite (MTM) is the most commonly used as reinforcing material in composite applications [25]. According to the nature of the clay, the nanoplatelet elastic modulus ranges from 50 to 180 GPa [17]. MTM nanoplatelets consist of ~1 nm thick aluminosilicate layers stacked together to form thicker multilayer (700 nm–10 µm) plate-like nanoparticles with a very high aspect ratio. They exhibit low thermal and electrical conductivities (in the order of mS/m), thanks to the porous nature of clay minerals, making them a good candidate for thermal barrier and flame-retardant applications [18].

### 1.3. Technologies Enabling Industry Applications

By combining the knowledge of biological materials with processing techniques, synthetic materials with remarkable mechanical and functional properties can be designed. Several methods have been used for the production of nacre-like materials, capable of reproducing the hierarchical well-organized microstructure of nacre, with lamellar nanoparticles aligned in the longitudinal direction and bonded together by a thin matrix layer 1 [2,26].

Generally, production methods follow two different approaches: top-down and bottom-up. For a definition, top-down methods go from a general to a specific level, while bottom-up methods begin at a specific level and move to the general one. More specifically, in top-down technologies the material is fabricated starting from a mixture of nanoparticles and polymer, which is assembled in such a way as to ensure a layered structure (Figure 1a). Conversely, in bottom-up technologies, the material is specified in great detail, by alternatively arranging the two phases and building up a layered structure (Figure 1b).

In general, bottom-up technologies are more able to fine-tune the nanometric alternance of the two phases [27] but remain mostly confined to the lab. Differently, top-down manufacturing processes are suitable for an industrial scale-up but are still far from obtaining the expected material architecture.

However, the final assembly of the bulk material is governed by different driving forces, which are involved during the fabrication process both in bottom-up and top-down technologies. Suter et al. [28] found that the formation of highly-ordered brick-and-mortar structures depends on the interaction forces between the two phases. If the flakes are relatively uncharged the bonds between the flakes and the polymer drive the self-assembly to the final highly-ordered structure. This means that the properties of a macroscopic bulk material strongly depend on the interaction at atomic levels, including van der Waals (vdW) force, hydrogen, ionic and covalent bonds and in most cases, on their synergetic effect. Of all the assembly interactions, vdW bonds and π–π interactions are the weakest and covalent bonds are the strongest, while hydrogen and ionic bonds are in between them [29,30,31].

Thus, to guarantee optimal mechanical performances, the interface mechanisms should be accurately adjusted to allow the optimal stress transfer from one phase to the other. In addition, particular attention has to be paid to the nature of the matrix, which can affect the interface and thus the overall behaviour of the composite.

## 2. Mechanical Performances—Experimental Observation of Literature Data

Experimental evidence demonstrated that the mechanical performances of brick-and-mortar materials depend on filler and matrix nature and on their interactions. The choice of the two phases and their compatibility is fundamental for achieving the desired performances.

Figure 2 shows an Ashby plot of strength and elastic modulus of artificial nacre with bricks of various nature. Graphene oxide (GO) nanoparticles, and in particular reduced graphene oxide (RGO), guarantee the best performances in terms of strength of the material, while exhibiting elastic moduli in the range of 15–40 GPa and 3–15 GPa, respectively. On the other hand, composites made with ceramic bricks, such as montmorillonite (MTM), show low values of strength but high elastic moduli (10–35 GPa). Finally, composites reinforced with graphene nanoplatelets (GNPs) exhibit the lowest values of strength but discrete elastic moduli in the range of 20–30 GPa. The low number of points indicates that they are not widely used as reinforcement in brick-and-mortar composites because of the difficulty of the nanoplatelets to be well dispersed in polymers.

Furthermore, these ranges are wide and depend on different factors, such as volumetric filler content, range of motion of the polymeric chains and the filler/matrix compatibility. The best performances can be achieved by improving the compatibility between nanoparticles and the polymers and their interactions, for example by chemically functionalizing the nanoplatelets or improving crosslinking. In fact, in Figure 2 the highest values of elastic moduli are achieved when the chemical affinity between the two phases is improved, for example by using glutaraldehyde (GA) [32,33], boric acid (BA) [34] or water (H_2_O) [35], or by functionalizing GO with polydopamine (P-GO) [36].

**Figure 2 nanomaterials-12-01359-f002:**
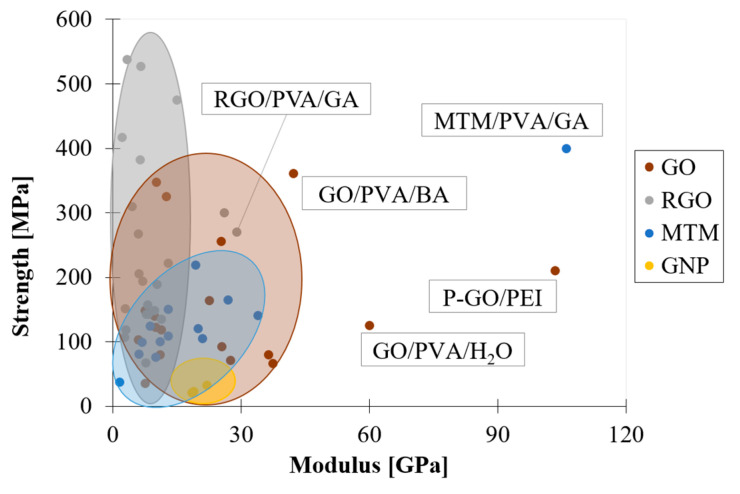
Ashby plot of strength vs. modulus of brick-and-mortar composite with bricks of various nature: GO [34,35,36,37,38,39,40,41,42,43,44,45,46,47]; RGO [14,33,38,41,48,49,50,51,52,53,54,55,56,57]; MTM [32,58,59,60,61,62,63]; GNP [16,64,65].

The mechanical properties of composites reinforced with lamellar nanoparticles usually depend on alignment and interfacial properties. At high filler content, the bricks tend to be extremely oriented in-plane because the available volume for nanoparticle rotations or displacement is limited, with a consequent waviness reduction and alignment improvement. Thus, for nacre-like materials, the mechanical performances mainly depend on the stress transfer between filler and matrix. The efficiency of reinforcement is strictly related to the interfacial properties, i.e., to the chemical affinity between the two phases, to the matrix wettability and to the molecular interactions that occur between adjacent nanoplatelets.

The purpose of this review is to give insight into the stress transfer mechanism in high filler content composites and to describe the parameters that influence the efficiency of stress transfer and the strategies to improve it (Figure 3).

### 2.1. Influence of Filler Content

From a critical analysis of the mechanical behaviour of composites with nano-lamellar reinforcement at relatively high filler content, it emerges that the elastic modulus of these systems drops after a critical concentration deviating from the expected behaviour, which dictates that the higher the filler content the higher the macroscopic elastic modulus. This unusual behaviour is reported by many authors in the literature.

Wu and Dzral [16] fabricated a self-standing graphite paper consisting of graphite nanoplatelets by vacuum filtration and impregnated it with a different amount of polyetherimide (PEI). By adding 30 wt% of polymer the tensile modulus reaches 22 GPa but then drops with further increase of PEI content. Likewise, Li et al. [65] produced high content GNP polyetherimide (PEI) paper by filtration and hot-press. They investigated the tensile properties of various filler contents and showed a maximum in the elastic modulus for a filler content of 60 wt%. This behaviour was also observed in a composite prepared with graphene oxide and alumina. In fact, highly-ordered GO/PVA papers, with high nanofiller concentrations, prepared using vacuum-assisted self-assembly technique, showed a reduction in the elastic modulus from 36.4 GPa to 27.6 GPa for 50 wt% to 75 wt% filler content, respectively [45]. In the same way, GO/PVA paper fabricated through a simple solution-casting method showed a maximum at 80 wt% of GO with an elastic modulus of 11.4 GPa [46]. Tian et al. [36] fabricated graphene-based paper via vacuum filtration with a low amount of polymer (<45 wt%). They used GO doped with polydopamine (PGO) to prepare PGO/PEI papers and functionalized them by a crosslinking reaction. The mechanical properties increased with the addition of PEI, showing a maximum strength (209.9 MPa) and modulus (103.4 GPa) for PEI loading of 14.7 wt%. GO/Thermoplastic polyurethane (TPU) films, with nacre-like laminated structure, are fabricated via solution casting with different matrix contents, achieving the best performances for 20 wt% of the polymer [42]. In addition, in alumina/chitosan nacre-like composites fabricated with the doctor blading technique, the mechanical properties are found to drop at high filler content [66]. In fact, at very high tablet concentrations (>50 vol%), the material failed in a brittle fashion, due to a misalignment of the tablets and the inability of the polymer to infiltrate the open spaces between tablets. Cao et al. [54] fabricated an RGO/PVA composite with an ordered layered structure and with a high level of molecular coupling between RGO sheets and PVA molecules. PVA chains strongly couple with RGO sheets at the molecular level and the neighbouring RGO sheets are linked by PVA molecules through hydrogen bonds and the C–C covalent bonds in the PVA polymer chains, forming a stronger multi-connected bridge as a continuous load transfer pathway. The films display extremely high strength, and Young’s modulus and the optimal content of PVA was found to be 70 vol%. Wang et al. [62] fabricated Montmorillonite/poly(vinyl alcohol) (MTM/PVA) nanocomposites spanning the complete range of MMT contents (0–100 wt%) by the simple evaporation-induced assembly. In the range of 30–70 wt%, the nanocomposites show a nacre-like layered structure with alternating MTM platelets and PVA layers. Composites reached the maximum value in terms of elastic modulus and strength for MTM content of 70 wt%, then for higher filler content, the layered structure was transformed into ”tactoids”, which are responsible for the deterioration of the mechanical properties. This suggests that partial MTM platelets are restacked and form “tactoids”, probably because the PVA layer is too little to fully cover all the MTM platelets.

This evidence demonstrates that the mechanical performances of composites with high contents of 2D nanoparticles are regulated by the efficacy of the thin polymeric layer to transfer the load via the shear transfer mechanism. In particular, the efficiency of reinforcement, *η*, is defined according to the modified rule of mixture and is based on the elastic modulus of the composite, *E_c_*:(1)$$\eta =\frac{E_{c}-E_{m}(1-v_{f})}{v_{f}E_{f}}$$

Figure 4 shows the unusual behaviour of this class of material for which with the increasing filler content the efficiency drops below one, indicating that the stress transfer between filler and matrix is poor. As a consequence, the elastic modulus deviates from the rule of mixture, exhibiting a decreasing behaviour (Figure 4b).

The drop of elastic modulus at high volume fraction is due to a bad interaction between the two phases and specifically to bad compatibility. This compatibility is governed by the chemical interactions between nanoplatelets and the matrix. In particular, Cilento et al. [68] observed that in GNP/Epoxy films, at high filler content (*v_f_* > 50 vol%) there are wettability issues: the polymer is not able to wet the entire nanoplatelet surface and accumulates into small pockets or droplets, which are smaller than the nanoplatelet surface. Consequently, there is a reduction of available effective length for stress transfer between the matrix and the nanoparticle, which is responsible for the decrease of reinforcement efficiency in these composites.

### 2.2. Influence of Matrix—Effect of Matrix Molecular Weight

In particular, the matrix choice can be discriminatory for the optimal mechanical performance of the material. The higher the molecular mobility of the polymer and the capacity to intercalate between nanoplatelets, the better the stress transfer at the interface and thus the performance of the material even at high filler content. Short polymer chains are able to diffuse between nanoparticles during assembly, while very long polymer chains’ ability to navigate around the layered nanosheets is more limited [45,69]. Evidence of this behaviour can be found by comparing the volumetric filler fraction at which the drop of efficiency occurs and the matrix molecular weight (Figure 5). For high molecular weights (150–300 kDa), the drop of efficiency occurs for very low volumetric filler content (20–30 vol%), whereas for low molecular weight (<100 kDa), the drop occurs for filler content greater than 40 vol%.

This indicates that the matrix molecular mobility affects the efficiency of reinforcement, which decreases as the molecular weight increases. The dependence of the efficiency of reinforcement from the matrix molecular weight is also shown in Figure 6 for a system reinforced with different nanoplatelets, especially in the case of MTM and GO. In the case of RGO, there is little evidence of this behaviour since the efficiency is very low (<10%) (Figure 6c).

This phenomenon has been found by other authors in the literature. Podsiadlo et al. [70] observed that the polymer flexibility directly affects the stress dissipation and load transfer from the organic matrix to the inorganic nanoscale component. In particular, they compared the mechanical response of B&M composites reinforced with MTM, observing that the elastic modulus of MTM/poly(diallyl dimethylammonium chloride) (PDDA) is greater than that of MTM/Chitosan (CS), even though the elastic modulus of CS is higher than that of PDDA. This can be explained by the high rigidity of CS and the poor MTM/CS interactions, which contribute to lowering the mechanical properties. The CS chains cannot find an optimal conformation on the surface of the MTM, due to lack of flexibility, which instead is possible for PDDA, where the strength of adhesion is about four times higher than that of MTM/CS and the attraction energies include electrostatic attraction, hydrogen bonding, and van der Waals forces.

In addition, Putz et al. [45] investigated the hydrophilicity and hydrophobicity of polymers, fabricating GO films with poly(vinyl alcohol) (PVA) and poly(methyl methacrylate) (PMMA). They observed poor interactions between GO sheets with hydrophobic PMMA, which limits improvement in stiffness at high nanofiller concentrations.

The importance of matrix molecular weight in the fabrication process of this class of material can also be understood by comparing the values of the theoretical and measured thicknesses of the matrix layer at different filler content. According to the representative volume element (RVE) of Figure 7, matrix thickness decreases with increasing filler content and is computed with Equation (2).
(2)$$t_{m}=\frac{1-v_{f}}{v_{f}}\cdot t_{NP}$$

In reality, from experimental data reported for GO/PVA [45], PGO/PVA [36] and RGO/PVA [54] systems, this value deviates from the theoretical one starting from the volumetric filler fractions which correspond to the drop of mechanical properties, as shown in Figure 8. At high filler content, the thickness of the matrix layer is higher than the theoretical value, meaning that the matrix does not spread as a continuous thin layer. When the matrix content is very low, the polymer thickness should be very small (<1 nm), but it could be incompatible with the mobility of the polymeric chains, especially when the matrix molecular weight is high. Putz et al. [45] found that at high filler content the polymer thickness is almost twice as much in composites with PMMA with respect to PVA due to its higher molecular weight.

Consequently, the polymer reaches a minimum thickness, which depends on matrix molecular weight, after which it does not decrease anymore. According to Figure 7, in order to guarantee the balance between the volumes, at high filler content, when the matrix thickness reaches a critical value, partial uncovering of nanoparticle surface occurs [68].

In other words, the matrix thickness must be compatible with the gyration radius of the polymer [71], otherwise dewetting of the nanoparticle surface can occur. This condition prevents full covering of the nanoparticle and compromises the performance of the brick-and-mortar material. Parameters that influence the wettability of thin polymeric films include molecular weight, temperature, film thickness, substrate interactions or their combinations [72,73]. These negatively affect the stress transfer and thus the composites’ mechanical behaviour, leading to a drop of strength and stiffness, especially at very high filler content (>70 wt%).

From Figure 6a, it emerges that the best mechanical performances are achieved when the nanoparticles are functionalized. In this respect, Podsiadlo et al. [32] and Walther et al. [61] used glutaraldehyde (GA) to improve PVA crosslinking to connect nanoclays. Despite all this, the improvement of mechanical properties is more significant in samples prepared by Podsiadlo et al., where the efficiency is maximum, rather than those prepared by Walther et al. This difference is attributed to the manufacturing process. In fact, Podsiadlo et al. used the LbL technique to produce MTM/PVA composites, dipping the film in a GA solution gradually every 0.05 μm for about 30 steps. On the contrary, Walther et al. produced a 200 μm MTM/PVA film with doctor blading and directly immersed it in the GA solution. The excellent barrier properties provided by the well-aligned microstructure of the material make it difficult for GA to penetrate in the bulk, especially in samples prepared by Walther et al. Thus, the treatment performed by Podsiadlo et al. is much more rigorous than the second one and involves a higher volume, justifying the significant improvement of the mechanical properties.

### 2.3. Filler/Matrix Compatibility—Chemical Bonding

As shown so far, the best mechanical performances are achieved when the chemical interactions between the two phases are improved [29]. It was found in the literature that functionalizations with glutheraldeyde (GA) [32], boric acid (BA) [74], water (H_2_O) or polydopamine are very efficient, as can be seen in Figure 2. This effect can also be observed in GO films without a binder. Figure 9 reports the efficiency of reinforcement for GO films with chemical bonds of various strength: covalent bond using BA [74], hydrogen bond using H_2_O [45] and ionic bond using Al^3+^, Mg^2+^ ions [75] and GA [76]. As expected, BA crosslinking creates strong covalent bonding, which significantly improves the elastic modulus by 240% with respect to not functionalized GO films. On the other hand, functionalization with H_2_O slightly increases the modulus by 25%, while ions lower the mechanical properties of the material due to the increase of the spacing between nanosheets.

However, the strength of chemical bonds also depends on the nature of the filler. For example, graphitic nanoplatelets, such as GO, RGO, GNP, are very different from a chemical point of view. Specifically, GO contains oxygen atoms on the surface, which interact with the polymeric matrix establishing covalent, ionic or hydrogen bonds. After oxidation, the oxygen content of RGO reduces significantly and with it the possibility to create strong chemical bonds with the polymeric matrix. As a consequence, the mechanical performances are very poor, as demonstrated previously (Figure 6c). The same discussion can be carried out for GNPs, which can interact only with weak vdW bonds and π–π interactions.

Consequently, the efficiency of reinforcement sharply decreases with increasing carbon/oxygen ratio (C/O), as shown in Figure 10. Typical values of the C/O ratio lie around 2 for GO and increases in the case of RGO being greater than 4 [77].

Suter et al. investigated the influence of C/O ratio on the morphology of GO/PVA structures [28]. They found that GO self-assembly can be controlled by changing the degree of oxidation, varying from fully aggregated to intercalated assemblies with polymer layers between sheets. The architecture varies according to the degree of oxidation. In the case of zero oxidation (RGO) it comes predominantly aggregated, with no polymer resident between the flakes and with the majority of flakes directly interacting with each other via attractive van der Waals interactions. When increasing the degree of oxidation, systems preferentially self-assemble, with a tendency toward forming intercalated morphologies, with GO flakes lying directly above each other and with the polymer between the flakes. The high oxidation degree allows a highly attractive interaction between PVA molecules with the hydroxyl groups on GO flakes, leading to a very dense layer of immobile polymer on each flake [35].

### 2.4. Interfacial Shear Strength

The efficient matrix–nanoplatelet stress transfer is essential to take advantage of the very high Young’s modulus and strength of the reinforcement. To assess the efficiency of reinforcement in nanocomposites, the interfacial property, which includes wetting, stress transfer and adhesion, should be thoroughly examined.

The experimental evaluation of the interfacial shear strength (IFSS) in a direct way is a challenging task, due to the technical difficulties involved in the manipulation of nanoscale objects. In carbon nanomaterials, the interfacial mechanics can be investigated by integrating scanning probe methods with spectroscopic techniques, such as tip-enhanced atomic force microscopy (AFM) [78] and Raman spectroscopy.

Experimental techniques able to quantitatively evaluate the IFSS at the nanoscale level are still demanding. Few works report pullout tests on carbon nanotubes (CNT) performed with AFM [79,80], estimating an average interfacial strength of 150 MPa [78]. In the case of nanoplatelets, conducting an experiment at the nanoscale is much more difficult, if not sometimes impossible. AFM examination of the interphase region requires a specific configuration to expose the interphase to the tip. Kraunbuel et al. [81] sandwiched a graphene nanoplatelet between two materials to measure the interfacial forces by AFM. Similarly Liu et al. [82] measured the IFSS in a macroscopic experiment by spraying graphene nanoplatelets on a PMMA substrate. The IFSS was measured by the stretching test, and the interface separation occurred as a result of the interfacial gliding of the two surfaces in contact.

Computational and analytical methods could be used to obtain information about the interfacial properties between the nanoplatelet and the polymer. Several studies performed molecular dynamics (MD) simulations to study the overall stress–strain response of GNP/polymer nanocomposites. Jang et al. [83] estimated an IFSS via a pullout energy method in a GNP/vinyl ester (VE) nanocomposite of 141 MPa, while Safaei et al. [84] found that the IFSS in GNP/HDPE nanocomposites is 108 MPa.

Specifically, the interfacial shear strength depends on both the polymer’s chemistry and the surface’s chemical makeup. The influence of oxidized carbon surfaces on interfacial properties is a significant parameter since the number of interfacial crosslinks improves the interfacial shear strength. In GO, the oxidation degree promotes the formation of hydrogen bonds, increasing the interfacial strength [85]. MD simulations also demonstrate that crosslink agents with strong interactions can be a promising strategy to strengthen and toughen nanocomposites with B&M architecture. In fact, by functionalizing nanoplatelets, the interlayer load transfer is improved thanks to the interlayer crosslinking [86].

To overcome the issues associated with the direct evaluation of IFSS, indirect methods, such as Raman spectroscopy, are widely used [87]. Raman spectroscopy gives information about the quality of stress transfer between the matrix and the reinforcement [88]. It is a valuable tool for understanding the relationship between macroscopic deformation and the deformation mechanism at the molecular or microstructural level. When the material is subjected to an external load, the nanoparticles and their chemical bonds are stressed and the interatomic distance changes, resulting in a translation of the spectrum peaks [89,90]. Thus, by monitoring the wavenumber shift of the Raman bands when a macroscopic stress/strain is applied (Raman shift rate), it is possible to identify the stress level within the nanoparticles and thus the capability of the matrix to transfer load [91,92].

The sensitiveness of frequencies/wavenumbers of the Raman bands with the applied external solicitation depends on the nanofiller [93,94,95,96]. Graphitic nanoparticles are particularly sensitive to the applied external solicitation [97,98,99].

The Raman shift rate can be proportionally correlated with the effective Young’s modulus of the nanoplatelet (*E_eff_*). In carbon-based materials, the effective modulus can be estimated from the slope of the Raman band position against the strain [29], as follows:(3)$$E_{eff}=E_{NP}\frac{d\omega /d\varepsilon }{{\left(d\omega /d\varepsilon \right)}_{\mathrm{ref}}}$$
where (d*ω*/d*ε*) is the measured Raman shift rate of the fundamental peak, *E_NP_* is the modulus of the nanoplatelet and (d*ω*/d*ε*)_ref_ is the rate of the peak shift for the isolated nanoplatelet.

## 3. Analytical Models for the Prediction of Mechanical Properties

At high filler content, the material architecture involves complex deformation mechanisms. The coexistence of a soft domain (polymeric layer) and a hard domain (bricks) could affect molecular mobility, leading to an increase in ductility and energy dissipation.

Actual biological and engineering structures, such as nacre, display spatial variations in overlap lengths, with different distributions which can be relatively narrow, as in the case of columnar nacre, or very wide and even uniform, as in the case of sheet nacre.

However, modelling these complex microstructures with a single unit cell gives a reasonable representation of the mechanical response of the material, which is sufficiently reliable to examine trends and establish broad design guidelines.

Composites with brick-and-mortar architecture can be schematized as depicted in Figure 11. They consist of a uniform assembly of bricks glued together by a uniform matrix thin layer (mortar).

The behaviour in tension is described in Figure 12. At small strain both bricks and mortar move in the elastic field and the behaviour is linear. Then, four different failure mechanisms can occur [100]. If the brick is weaker than the matrix, there is an instant failure, which leads to a fragile behaviour of the structure. Otherwise, the matrix in the vertical interface yields, making the behaviour more ductile. In this case, for linear elastic matrices, the composite exhibits a pseudo-elastic behaviour and failure is attributed to the vertical junctions, whereas for matrices with elastic–plastic behaviour, yielding in the horizontal direction occurs, allowing sliding between bricks. The composite failure can be attributed to the mortar break either in the vertical junctions or in the horizontal interfaces. In both cases, large strains are reached with consequent sliding between bricks and the final pullout mechanism.

In modelling brick-and-mortar composites, it is assumed that the bricks behave elastically, while the mortar can have both elastic and elastoplastic behaviour. Small plane strain deformations are assumed with zero strain in the *z*-direction, and zero stress in the *y*-direction. In addition, under uniaxial deformation, the horizontal interfaces experience pure shear according to the relative displacements between adjacent bricks in different rows, while the vertical interfaces experience pure tension according to the relative displacements between adjacent bricks in the same rows. Finally, the horizontal mortar layer is considered very small, such that the shear strain is uniform. The representative volume elements (RVE) are accurately chosen according to the analytical model and are depicted in Table 3.

The mechanical performance of these composites is regulated by the efficacy of the thin polymeric layer to transfer load, via the shear transfer mechanism [101]. Thus, models [100,102,105,106] and design strategies [103,107,108] to predict the elastic modulus and strength of brick-and-mortar composites, based on the shear lag theory, have been proposed in the literature. These models assume perfect bonding between the two phases, efficient load transfer to the particles and the existence of a continuous uniform matrix layer between the bricks.

The shear lag theory was first introduced by Cox, who developed an accurate model for cylindrical fibre [101]. It considers a fibre completely embedded in a continuous solid matrix and loaded in the longitudinal direction. During loading, the fibre deforms the most in the central part and less on the edges, with consequent relative displacement between filler and matrix. Therefore, the stress transfer from the matrix to the fibre is governed by interfacial shear stresses, which are maximum at the edges.

Based on this theory, several micromechanical models for the prediction of the mechanical behaviour of discontinuous composites have been developed.

Kotha et al. [102] developed a similar model for the prediction of the mechanical properties of a 2D platelet embedded in a continuous matrix and loaded in tension along the direction of the tablets. Similar to Cox, the model assumes that the interface does not carry tensile stresses and that the axial stress in the matrix is constant. The RVE is symmetric, with overlapped platelets uniformly distributed and aligned in the direction of applied load and perfectly bonded by a uniform matrix layer. The matrix at the ends of the platelet is considered as an imaginary platelet with the same thickness of the brick and with the mechanical properties of the matrix that acts as a shear spring, transferring load from one platelet to the other.

Along this path, Barthelat [103] extended the model to the case of non-symmetric RVE, highlighting the impacts of the overlap length on the mechanical properties of the material. According to the structure of the nacre, the RVE is characterized by tablets with a well-defined arrangement and overlapped on a length *L*_0_. The tablets are assumed linear elastic and brittle, while the matrix is modelled as a linear elastic–perfectly plastic material. The deformation mechanism is of shear–tension–shear, where the tablets are loaded in tension and the matrix in shear. For this reason, the vertical junctions between the tablet are assumed to be empty. According to the shear lag theory, the tensile stress in the brick and the shear stress at the interface are not uniform. Focusing on an individual overlap region within the composite, the distribution of shear stresses along the interface is governed by a non-dimensional elastic shear transfer number *β*_0_, reported in Table 3.

When the interface is soft and/or the overlap ratio is small (*β*_0_ < 1), the shear stress is quasi-uniform along the overlap length. On the contrary, when *β*_0_ is large, the shear stresses become more concentrated at the overlap regions’ edges, accelerating failure. This is the case of low-efficiency structures, because in the central area of the overlap region, the interface does not carry any stress, and does not provide any contribution to the structural performance of the composite. Furthermore, for high brick concentration, variations in aspect ratio and overlap ratios have little effect on the overall modulus, leading to high modulus and resulting in more robust microstructures.

A more complex solution was found by Begley et al. [100], who included the vertical junctions in the model. A micromechanical analysis was developed for the uniaxial response of composites composed of elastic bricks bonded together with thin elastic–perfectly plastic layers. The unit cell contains two bricks separated horizontally and vertically by a mortar layer, interlocked by a distance *L*_0_. The model assumes that bricks are perfectly aligned, but with an arbitrary offset between rows, and that the vertical interfaces carry the load. The model is valid in the case of a very small mortar ratio and small mortar volume fraction, and the material failure corresponds to that of the interfaces.

Wei et al. [106] proposed an analytical model based on shear lag theory in both elastic and plastic regimes, capable of linking the mechanical properties of the constituents to the mechanical behaviour of the reinforcement and their geometric arrangement, and to the chemistries used in their lateral interactions. In particular, the model aims to define design guidelines and to predict the characteristic overlap length for the optimization of the mechanical performances of the material. The RVE consists of two tablets connected by the matrix with an overlap length *L*_0_. The system is subjected to a tensile load applied to the right end of the top tablet and fixed in the axial direction in the left end of the bottom tablet. The load transfer mechanism resembles that of the shear lag model, with maximum shear stress at both nanoplatelet ends and maximum normal stress in the centre of the platelet, like in common biological composites.

The effect of both geometry and matrix constitutive law on the macroscopic behaviour of the composite material was investigated by Pimenta and Robinson [105], who developed a model for brick-and-mortar structure with nonlinear matrix response. They highlighted the relevant influence of the platelet aspect ratio (AR) on composite behaviour. In fact, for thick configuration, the strength increases with the aspect ratio, the overall stress–strain curve resembles the matrix constitutive law and the distribution of shear stresses along the overlapping region is homogeneous. On the other hand, for slender configuration, the strength is independent of *AR*, and the stress–strain curve does not depend on the matrix behaviour but is governed by matrix fracture with a crack tip at the platelets’ ends and converges to a fracture criterion, as summarized in Table 4.

Furthermore, they highlighted the importance of brick thickness on the composite behaviour. In particular, in the case of thick platelets and strain hardening matrices, the composite has a ductile behaviour. In contrast, thin platelets delay the final failure and increase the strength of the composite.

However, all these models predict that the strength and the elastic modulus of the composites increase with the amount of filler, reaching the best performance, equal to those of the reinforcement, at very high filler content (90–95 wt%). In reality, for this class of material the mechanical performances at high filler content drop, diverging from the theoretical behaviour, because of the difficulty of building a continuous nanometric matrix film able to fully cover the nanoplatelet surface [109,110]

Cilento et al. developed an analytical model which describes the unusual behaviour (i.e., drop of modulus) that characterizes composites with lamellar reinforcement at high filler content. The model accounts for a non-uniform matrix distribution over the nanoplatelets. It considers that at low filler content nanoplatelets are fully covered, enabling the complete stress transfer between the two phases and therefore maximizing the efficiency of reinforcement. Increasing the filler content, the nanoplatelets are considered partially covered, since the matrix accumulates into small pockets or droplets, the height of which is compatible with the gyration radius of the polymer. In this case, the stress transfer is limited to a smaller area and the efficiency of reinforcement drops. This model takes into account the interfacial properties between the two phases in terms of nanoplatelet coverage since poor wetting of nanoplatelets inhibits the stress transfer mechanism. It incorporates two parameters that inherently capture the interfacial efficiency: (a) the minimum matrix thickness (*t_M_*), which depends on both the chemical affinity between the two phases and the matrix wettability, and (b) the cohesive parameter (*η^ii^*), which contains information about the molecular interactions that occur between adjacent nanoplatelets.

## 4. Discussion

Learning from nature has inspired the fabrication of novel artificial materials with outstanding performances. In the last decade, bioinspired research has led to numerous advances in materials science and has laid the foundation for a new class of biomimetic materials.

Among all of these materials, nacre drew attention thanks to its ability to reproduce on a large scale the mechanical performances of the nanometric reinforcement. This is due to the particular brick-and-mortar architecture composed of well-oriented thin laminae of aragonite (95 vol%), bonded with a low amount of organic material. According to this, by employing high-performant nanoparticles such as graphene as filler, it is possible to fabricate macroscopic materials with excellent performances that can be employed in different fields. However, achieving the desired performances is challenging, since the mechanical response of the material is influenced by many factors, such as the filler content, the matrix molecular mobility and the compatibility between the two phases. Most importantly, the properties of a macroscopic bulk material strongly depend on the interactions at the atomic level and on their synergetic effect.

However, contrary to what it is expected, nacre-like materials exhibit unusual behaviour at high filler content. The elastic modulus decreases with increasing filler content, deviating from the trend dictated by the rule of mixture. At high filler content, the efficiency of the reinforcement drops below one, indicating that the stress transfer between filler and matrix is poor. Indeed, the mechanical performances of composites with high content of 2D nanoparticles are regulated by the efficacy of the thin polymeric layer to transfer load via a shear transfer mechanism. When the matrix content is very low, the thickness of the polymeric layer should be very small (<1 nm). This can be incompatible with the mobility of the polymeric chains, especially when the matrix molecular weight is high. Consequently, dewetting of the nanoplatelet surface can occur at high filler content, causing a reduction of the effective length for stress transfer and impairing the reinforcement efficiency.

It follows that the matrix molecular mobility affects the efficiency of reinforcement. Polymers with high molecular weight are characterized by high rigidity, making it difficult for the polymeric chains to find an optimal conformation on the surface of the nanoplatelets. Therefore, the higher the polymer molecular mobility, the better its capacity to intercalate between nanoplatelets. In fact, it has been observed that for polymers with high molecular weights, the drop of efficiency occurs for very low volumetric filler content, whereas for polymers with low molecular weight, the drop occurs for higher filler content. Thus, the selection of the matrix is important for achieving high efficiency [111,112].

To sum up, it has been found that the best mechanical properties are achieved when the bonding and interactions between the two phases are strong and when the matrix molecular mobility is such that the polymer intercalates between nanoplatelets, covering their entire surface. This means that high performances and mechanical improvements need high interfacial attraction between fillers and the surrounding matrix that guarantee the load transfer.

To maximize the interfacial interactions, improvements of chemical bonds between the filler and matrix should be developed by functionalizing nanoparticles by introducing covalent, ionic or hydrogen bonding forces. In addition, other strategies aimed to improve the stress transfer efficiency can be employed, by promoting the wetting phenomenon and inhibiting the phenomenon of partial coverage. For example, increasing the compaction pressure during the manufacturing process can improve the stress transfer efficiency, thanks to a physical confinement of the polymer [68].

Basically, the filler/matrix compatibility is at the base of the optimal performance of the material. Both the chemical bonding and the matrix wettability should be carefully chosen in order to optimize the compatibility between the two phases. As a result, the interface optimization would lead to a shift of the drop point to higher filler content and the performances of the material would reach those of the reinforcement (Figure 13).

## 5. Conclusions

In this review, the authors revised the state of the art of nacre-like materials, highlighting the issues that come out in the fabrication of composites with high filler content and that reflect on their mechanical behaviour. The aim of the work is to give insight into the mechanisms which regulate the stress transfer efficiency in composites with B&M architecture, in order to lay the groundwork for the design of bioinspired materials with outstanding properties.

Although composites with brick-and-mortar architecture are promising from a theoretical point of view, in reality it is challenging to achieve the desired mechanical performances. The target is to assemble 2D nanoparticles in order to achieve a macroscopical material able to reproduce the unique properties of the 2D nanoparticle:$$v_{f}\to 1\;\;\;\;\;\;\;\Rightarrow \;\;\;\;\;\;E_{c}\approx E_{2D\;particle}$$

This condition can be achieved when the matrix arranges as a continuous nanometric film on the nanoplatelets’ surface and the complete stress transfer at the interface is guaranteed, according to the shear lag theory. However, the mechanical response of the material is influenced by several factors: volumetric filler content, matrix molecular mobility and compatibility between the two phases. Thus, the efficiency of reinforcement:−*Drops at high filler content*.Starting from a critical volumetric fraction, the elastic modulus B&M composites deviate from the expected behaviour dictated by the rule of mixture due to the partial coverage of the nanoplatelets at the nanoscopic level.−*Decreases as the molecular weight increases*.The higher the molecular mobility of the polymer and the capacity to intercalate between nanoplatelets, the better the stress transfer at the interface.−*Improves when a high interfacial attraction between nanoparticles and the surrounding matrix is guaranteed*.Strong chemical bonding and molecular interactions between nanoparticles and the polymer ensure self-assemblies with a tendency toward forming intercalated morphologies, with a stable layer of polymer between flakes.

## Figures and Tables

**Figure 1 nanomaterials-12-01359-f001:**
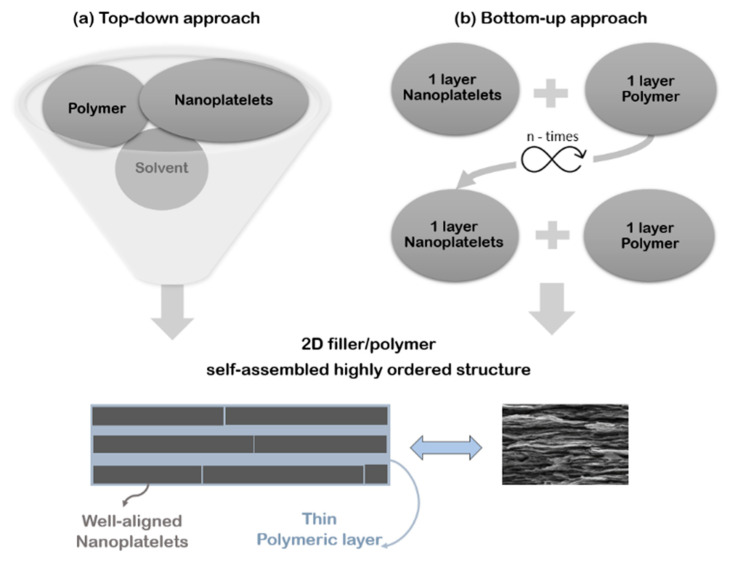
**Top-down** (**a**) and **bottom-up** (**b**) approaches for manufacturing of composites with B&M architecture.

**Figure 3 nanomaterials-12-01359-f003:**
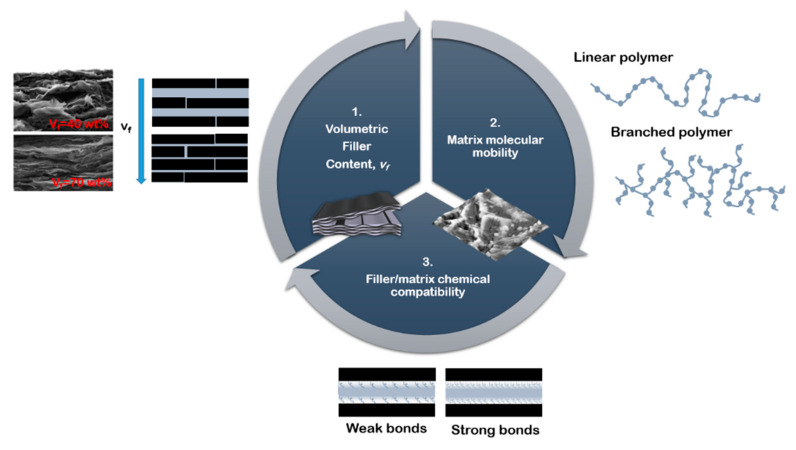
Parameters which influence the mechanical performances of composite with nano-lamellar reinforcement.

**Figure 4 nanomaterials-12-01359-f004:**
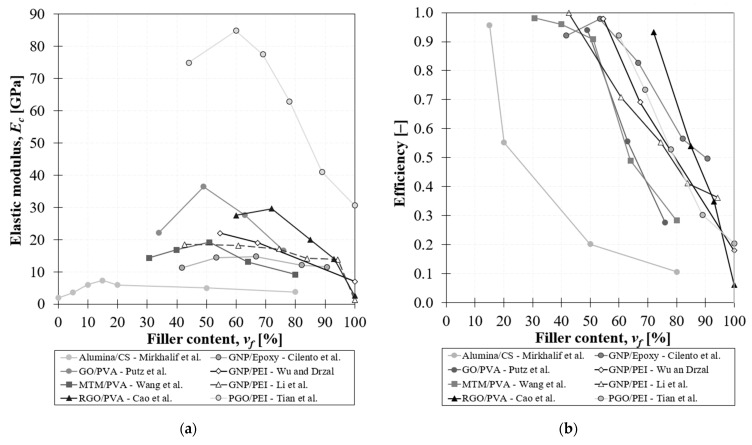
Drop of elastic modulus (**a**) and efficiency (**b**) at high filler content according to the state of art [16,36,42,45,46,54,62,65,66,67].

**Figure 5 nanomaterials-12-01359-f005:**
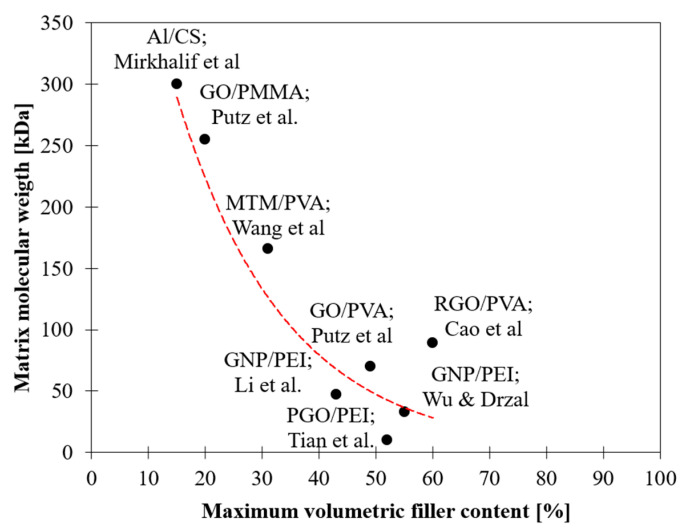
Dependence of drop off point on matrix molecular weight.

**Figure 6 nanomaterials-12-01359-f006:**
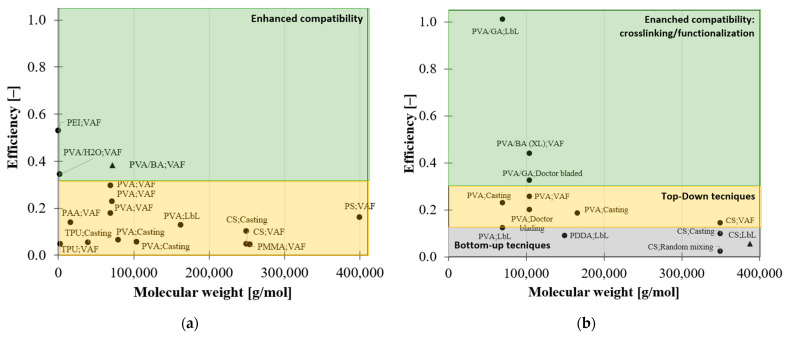
Efficiency of reinforcement vs. molecular weight for composite reinforced with: (**a**) MTM; (**b**) GO; (**c**) RGO.

**Figure 7 nanomaterials-12-01359-f007:**
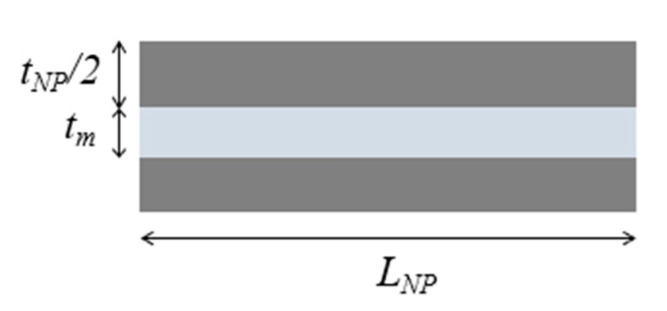
Representative volume element (RVE) of composites reinforced with lamellar nanofiller at high filler content.

**Figure 8 nanomaterials-12-01359-f008:**
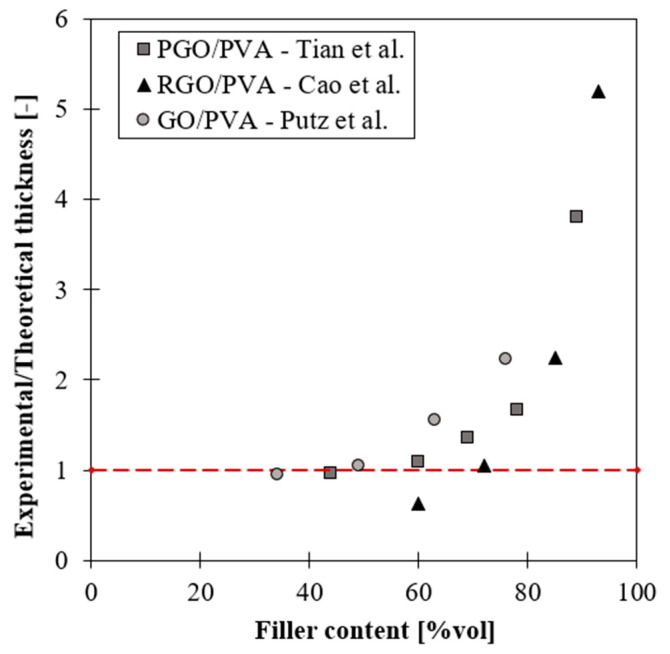
Comparison between experimental and theoretical matrix thickness in GO/PVA composites as function of filler content.

**Figure 9 nanomaterials-12-01359-f009:**
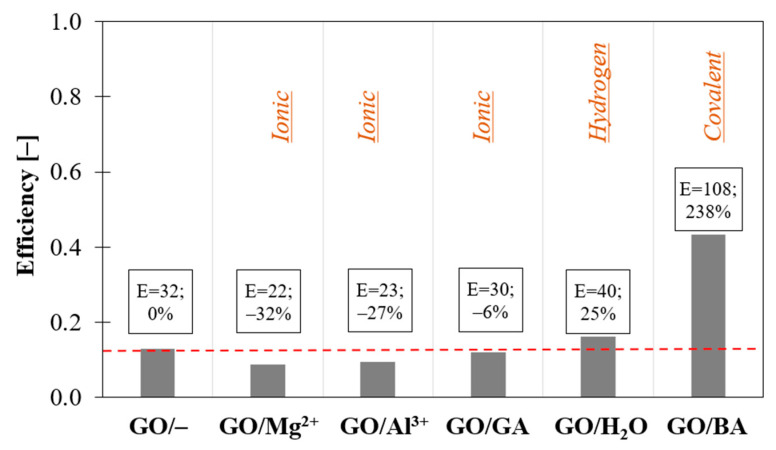
Effect of nanoparticle functionalization on the elastic modulus of GO films without binder.

**Figure 10 nanomaterials-12-01359-f010:**
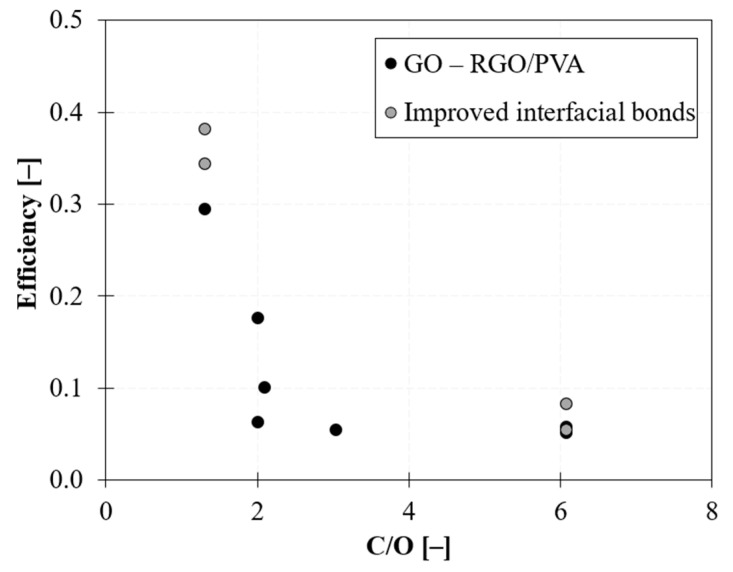
Influence of chemical interactions between filler and matrix: efficiency of reinforcement vs. C/O ratio in GO/PVA and RGO/PVA films.

**Figure 11 nanomaterials-12-01359-f011:**
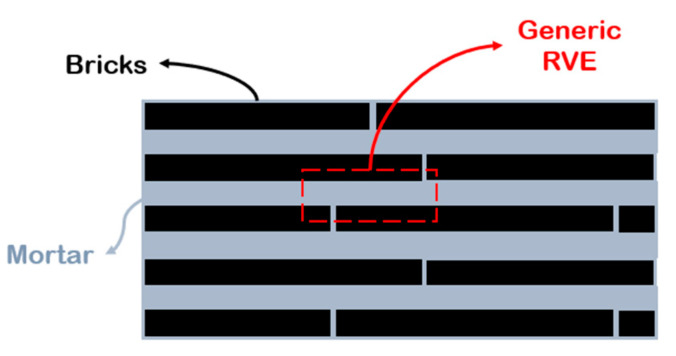
Schematic illustration of brick-and-mortar composites.

**Figure 12 nanomaterials-12-01359-f012:**
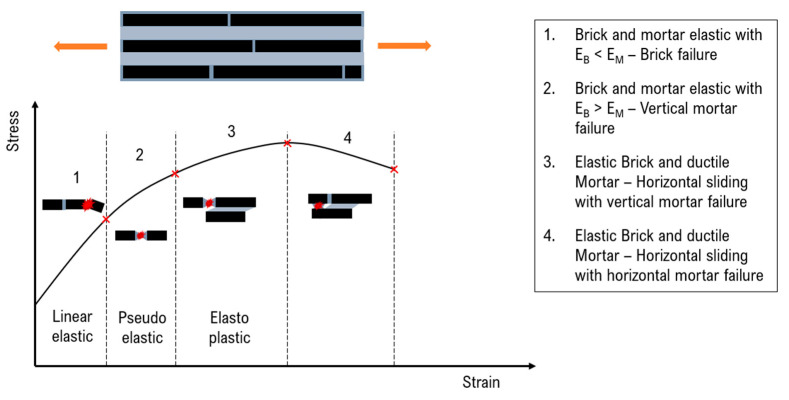
Stress–strain behaviour in tension of B&M composites.

**Figure 13 nanomaterials-12-01359-f013:**
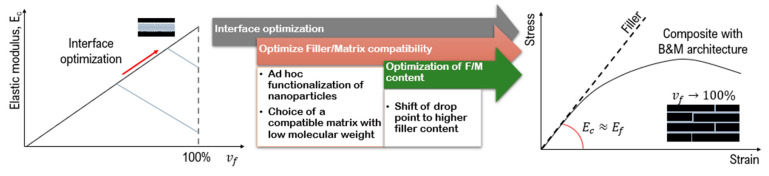
Strategies to optimize the mechanical properties of high filler content composites.

**Table 1 nanomaterials-12-01359-t001:** Mechanical and functional behaviour of composites with high content 2D nanofiller according to reinforcement nature.

Filler	Matrix Type	Composite Mechanical Behaviour	Composite Conductivities
Graphitic (GO, RGO, GNP, Pyrolytic Graphite…)	Brittle	Pseudo-elastic	Electrically Conductive in plane High ratio in plane/trough thickness thermal conductivity
	Ductile	Plastic	
Ceramic (MTM, Alumina, Silica…)	Brittle	Pseudo-elastic	Isolating
	Ductile	Plastic	

**Table 2 nanomaterials-12-01359-t002:** Comparison between 2D nanoplatelets employed as reinforcement in B&M composites.

Particle	Costs	Geometry	Elastic Modulus	In Plane—Therm. Cond.	Elec. Cond.
Graphene	EUR 200–300 per flake	Monolayer	1 TPa [9]	5000 W/mK [9]	10^7^–10^8^ S/m [9]
GO	2–5 layers 48 EUR/g	2–5 layer BET 420 m^2^/g	250 GPa [10]	72 W/mK with an oxidation degree of 0.35 [11]	270 S/m [12]
RGO	2–5 layers 68 EUR/g	2–5 layer BET 1562 m^2^/g	250–350 GPa [13]	670 W/mK with an oxidation degree of 0.05 [11]	4480 S/m [14]
GNP	6–10 EUR/g	>10 layer BET 30 m^2^/g	25–40 GPa [15]	300–470 W/mK [8]	2 × 10^6^ S/m [16]
MTM	<1 EUR/g	BET 750 m^2^ very high (nm × µm) aspect ratio	207 GPa [17]	16 W/m [18]	25 to 100 mS/m [18]

**Table 3 nanomaterials-12-01359-t003:** Analytical models for the prediction of composite’s elastic modulus.

Authors	RVE	Elastic Modulus
(a) Cox et al. [101]	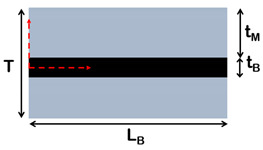	$E_{c}=E_{m}v_{m}+\eta \cdot E_{f}v_{f}$ $\eta =1-\frac{\tanh(AR_{B}\cdot n/2)}{AR_{B}\cdot n/2}$ $,\;\mathrm{with}\;AR_{B}=\frac{L_{B}}{t_{B}}$
(b) Kotha et al. [102]	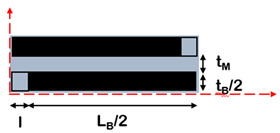	$E_{c}=E_{t}v_{f}\frac{1}{1+\coth\beta_{0}/\beta_{0}}$ $\beta_{0}=\rho_{0}\sqrt{\frac{G_{i}}{E_{t}}\frac{v_{f}}{1-v_{f}}};\;\rho_{0}=\frac{L_{0}}{t_{B}}$
(c) Barthelat et al. [103].	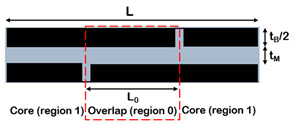	$E_{c}=\frac{E_{f}v_{f}}{1+\frac{\kappa }{\beta_{0}}\left[\coth\beta_{0}+\coth\left(\frac{1-\kappa }{\kappa }\beta_{0}\right)\right]}$ $\beta_{0}=\rho_{0}\sqrt{\frac{G_{i}}{E_{t}}\frac{v_{f}}{1-v_{f}}};\rho_{0}=\frac{L_{0}}{t_{B}}$
(d) Begley et al. [100]	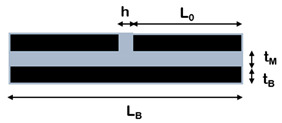	${\bar{E}}_{c}\approx \frac{2\kappa_{1}+\kappa_{2}}{2(1+\kappa_{1})+\kappa_{2}}$ $\begin{array}{l} \kappa_{2}=\kappa_{horizontal}=\sqrt{\frac{(1-v_{M}){\bar{E}}_{M}L_{B}^{2}}{2{\bar{E}}_{B}t_{M}t_{B}}}\\ \kappa_{1}=\kappa_{vertical}=\frac{{\bar{E}}_{M}L_{B}}{{\bar{E}}_{B}h} \end{array}$
(e) Wei et al. [104]	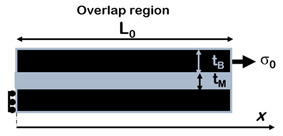	$E_{c}=E_{f}v_{f}\frac{1}{1+\frac{\coth\left(\lambda L_{0}/2\right)}{\lambda L_{0}/2}}$ $\lambda =\frac{1}{t_{B}}\sqrt{\frac{G_{m}}{E_{t}}\frac{v_{f}}{1-v_{f}}}$
(f) Cilento et al. [68]	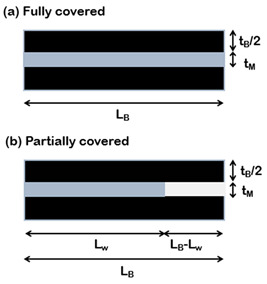	$E_{c}=E_{m}v_{m}+\eta \cdot E_{f}v_{f}$ $\eta =\frac{AR_{w}}{AR_{B}}\cdot \left(1-\frac{\tanh(AR_{w}\cdot n/2)}{AR_{w}\cdot n/2}\right)+\left(1-\frac{AR_{w}}{AR_{B}}\right)\cdot \eta^{\left(ii\right)}$ with $\begin{array}{l} AR_{w}=\frac{L_{w}}{t_{B}}=\frac{1-v_{f}}{v_{f}}\frac{L_{B}}{t_{M}};\\ AR_{B}=\frac{L_{B}}{t_{B}} \end{array}$

**Table 4 nanomaterials-12-01359-t004:** Dependence of mechanical behaviour on nanoplatelet aspect ratio.

	Thick Platelets (AR < 10)	Slender Platelets (AR > 30)
Strength	Increase with AR	Independent from AR
Stress–Strain curve	Resembles the matrix constitutive law	Does not depend on the matrix behaviour
Behaviour	Ductile (Yield criterion)	Fragile (Fracture criterion)
Fracture	-	Matrix fracture with crack tip at the platelets’ ends
Distribution of shear stresses	Homogeneous	-

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
