# Peer review of "Insights on Shear Transfer Efficiency in “Brick-and-Mortar” Composites Made of 2D Carbon Nanoparticles"

_nanomaterials, 2022, doi:10.3390/nano12081359_

Round 1

Reviewer 1 Report

This manuscript deals with overall insights and reviews on nano materials that have been popular in construction materials for the past 10 years.

In particular, it helps to organize at a glance by comparing 'analytical models for the prediction of composite's elastic modulus' through a common schematic diagram.

The overall content is scientific and logical.
However, in the case of Chapter 4, it is recommended to write it separately from the conclusion.
If the last chapter is too long, it may be difficult to see the purpose of this review paper at a glance.

It contains great content that can be published as it is, and thank you for your hard work.

Author Response

As the reviewer suggested, a new paragraph of conclusions has been added at the end of the manuscript, recapping and bringing together the main findings of the review.

Reviewer 2 Report

This review article focuses on the mechanical properties o polymer-filler composites. The material is interesting and comprehensive. However, there are currently issues with content, layout and approach that need to be fully resolved before the manuscript can be considered for publication.

Structure & Content:

  • The conclusion section is entirely missing from the article. It would be good to include as a means of recapping and bringing together all the material that has been covered.
  • There is a large focus on carbon-based filler materials in the manuscript, but nothing that is mentioned in either the title or abstract. It seems like an important detail to front-load.
  • There seems to be little/no coverage of compatibilisers between fillers and the host matrices. What kind of effect on the mechanical properties would such additives (and variations between them) have?

Language:

  • The writing could be much clearer – lots of spelling and grammatical errors are strewn throughout the manuscript (which is otherwise, relatively clear).
  • For example, the title includes the phrase ‘brick-and mortar’. This is a specialist phrase that is used within the sub-discipline of polymer composites. Thus, the phrase should be placed in quote marks and/or italicised. Otherwise, it could be very easily misunderstood (and understandably so), by a non-specialist as a topic that covers architectural building materials.
  • Likewise, the term needs to be properly and clearly defined in the first paragraph of the introduction. Especially as this is a review article, up-to-date and complete definitions and explanations need to be given.

Figures:

  • All figure labels are overly brief. It would be near-impossible for someone who has not otherwise read the paper to comprehend what is going on!
  • Other issues:
  • Fig 3: What is the general trend/relationship that would be expected for each of the parameters listed? It is insufficient to simply list the parameters, as has currently been done. Where is the value?
  • Fig 4b: There seems to be data cut-off, according to the y-axis. Please address this issue.

References:

  • There are a few issues,
    • g.; missing journal titles (37, 52, 53)
    • g.; issues with journal titles (49)

Author Response

This review article focuses on the mechanical properties o polymer-filler composites. The material is interesting and comprehensive. However, there are currently issues with content, layout and approach that need to be fully resolved before the manuscript can be considered for publication.

Structure & Content:

  1. “The conclusion section is entirely missing from the article. It would be good to include as a means of recapping and bringing together all the material that has been covered.

A conclusion section has been added. In order to better explain the aim of the review, a bullet point of the main findings is presented.

  1. “There is a large focus on carbon-based filler materials in the manuscript, but nothing that is mentioned in either the title or abstract. It seems like an important detail to front-load.“

Thanks to the reviewer for the suggestion. The title as been modified, as follow: “Insights on shear transfer efficiency in “brick-and-mortar” composites made of 2D carbon nanoparticles”.

Also, the abstract has been updated, as follow:

“…The purpose of this review is to give an insight into stress transfer mechanism in high filler content composites reinforced with 2D carbon nanoparticles and to describe the parameters that influence the efficiency of stress transfer and the strategies to improve it.“

  1. There seems to be little/no coverage of compatibilisers between fillers and the host matrices. What kind of effect on the mechanical properties would such additives (and variations between them) have?

We are grateful to the reviewer remark, which encouraged us to investigate this other possibility. We suppose that all the strategies aimed to improve the compatibility between the two phases could solve the issues associated to the partial coverage of nanoplatelets at high filler content.

Actually, the idea of using compatibilizers was not explored yet. We think that it could be an excellent strategy to improve the compatibility between the filler and the matrix in the same way as the functionalization of nanoplatelets. In literature, is well documented the use of compatibilizers to improve both nanoparticle distribution and interfacial interactions between the inorganic nanofillers and the polymeric matrix [1–5] Even if this strategy is appealing, there are not enough studies regarding the application in nacre-like materials, which is the focus of this paper.

Language:

  1. The writing could be much clearer – lots of spelling and grammatical errors are strewn throughout the manuscript (which is otherwise, relatively clear).

The paper has been revised. Minor grammatical errors have been fixed and some non-clear sentences have been rephrased. The modified text is highlighted.

  1. For example, the title includes the phrase ‘brick-and mortar’. This is a specialist phrase that is used within the sub-discipline of polymer composites. Thus, the phrase should be placed in quote marks and/or italicised. Otherwise, it could be very easily misunderstood (and understandably so), by a non-specialist as a topic that covers architectural building materials.

The phrase has been placed in quote marks in the title, according to the reviewer’s suggestion, and the title has been modified.

  1. Likewise, the term needs to be properly and clearly defined in the first paragraph of the introduction. Especially as this is a review article, up-to-date and complete definitions and explanations need to be given.

The introduction paragraph reports the brick-and-mortar architecture of nacre and the corresponding synthetic composites. The section “Mimicking Nature: nacre-inspired materials” describes the concept of B&M, which is depicted in Figure 1.

Figures:

  1. All figure labels are overly brief. It would be near-impossible for someone who has not otherwise read the paper to comprehend what is going on!

The figure labels have been revised.

  1. Other issues: Fig 3: What is the general trend/relationship that would be expected for each of the parameters listed? It is insufficient to simply list the parameters, as has currently been done. Where is the value?

The aim of Figure 3 is to identify main parameters contributing to the excellent mechanical properties related to B&M materials.

The target would be to assembly 2D nanoparticles in order to achieve a macroscopical material able to reproduce the outstanding properties of the nanoparticle, mimicking nacre:

This condition can be achieved when the matrix arranges as a continuous nanometric film on nanoplatelets surface. In fact, according to the shear lag theory applied to lamellar nanoparticles, the complete stress transfer at the interface is guaranteed and the efficiency of reinforcement is equal to 1.

The strategies, which enable maximum reinforcement, rely on enhancing nanoplatelet/matrix interface and using polymers with low molecular weight (radius of gyration is compatible with the theoretical matrix thickness of Eq. (2)).

  1. Fig 4b: There seems to be data cut-off, according to the y-axis. Please address this issue.

The plot has been adjusted according to the reviewer comment.

References:

  1. There are a few issues, g.; missing journal titles (37, 52, 53) g.; issues with journal titles (49)

References have been updated.

REFERENCES

[1]          O. Bas, F. Hanßke, J. Lim, A. Ravichandran, E. Kemnitz, S.H. Teoh, Di.W. Hutmacher, H.G. Börner, Tuning mechanical reinforcement and bioactivity of 3D printed ternary nanocomposites by interfacial peptide-polymer conjugates, Biofabrication. 11 (2019). https://doi.org/10.1088/1758-5090/aafec8.

[2]          A.U. Chaudhry, V. Mittal, High-Density Polyethylene Nanocomposites Using Masterbatches of Chlorinated Polyethylene/Graphene Oxide, Polym. Eng. Sci. (2013) 1–10. https://doi.org/10.1002/pen.

[3]          V. Mittal, A.U. Chaudhry, Effect of amphiphilic compatibilizers on the filler dispersion and properties of polyethylene - Thermally reduced graphene nanocomposites, J. Appl. Polym. Sci. 132 (2015) 1–11. https://doi.org/10.1002/app.42484.

[4]          Y. Sun, X. Fan, X. Lu, C. He, Overcome the Conflict between Strength and Toughness in Poly(lactide) Nanocomposites through Tailoring Matrix–Filler Interface, Macromol. Rapid Commun. 40 (2019) 1–6. https://doi.org/10.1002/marc.201800047.

[5]          S. Ghosh, G. Otorgust, A. Idelevich, O. Regev, I. Lapsker, D.Y. Lewitus, A. Zak, Reinforcement of poly (methyl methacrylate) by WS2 nanotubes towards antiballistic applications, Compos. Sci. Technol. 207 (2021) 108736. https://doi.org/10.1016/j.compscitech.2021.108736.

Round 2

Reviewer 2 Report

The changes address the major highlighted issues. Apart from a final readthrough to address minor errors, the manuscript should be ready for publication. 

Author Response

The paper has been updated after a final readthrough.
